# Assessment of Childhood Stunting Prevalence over Time and Risk Factors of Stunting in the Healthy Village Programme Areas in Bangladesh

**DOI:** 10.3390/children11060650

**Published:** 2024-05-28

**Authors:** May Phyu Sin, Birger C. Forsberg, Stefan Swartling Peterson, Tobias Alfvén

**Affiliations:** 1Department of Orthopedics, Lund University, 221 85 Lund, Sweden; may_phyu.sin@med.lu.se; 2Department of Global Public Health, Karolinska Institutet, 171 77 Stockholm, Sweden; stefan.swartling.peterson@ki.se (S.S.P.); tobias.alfven@ki.se (T.A.)

**Keywords:** stunting, growth monitoring, WASH, under-five children, prevalence, risk factors, Bangladesh

## Abstract

Childhood stunting is a significant public health concern in Bangladesh. This study analysed the data from the Healthy Village programme, which aims to address childhood stunting in southern coastal Bangladesh. The aim was to assess childhood stunting prevalence over time and explore the risk factors in the programme areas. A cross-sectional, secondary data analysis was conducted for point-prevalence estimates of stunting from 2018 to 2021, including 132,038 anthropometric measurements of under-five children. Multivariate logistic regression analyses were conducted for risk factor analysis (*n* = 20,174). Stunting prevalence decreased from 51% in 2018 to 25% in 2021. The risk of stunting increased in hardcore poor (aOR: 1.46, 95% CI: 1.27, 1.68) and poor (aOR: 1.50, 95% CI: 1.33, 1.70) versus rich households, children with mothers who were illiterate (aOR: 1.25, 95% CI: 1.09, 1.44) and could read and write (aOR: 1.35, 95% CI: 1.16, 1.56) versus mothers with higher education, and children aged 1–2 years compared with children under one year (aOR: 1.32, 95% CI: 1.20, 1.45). The stunting rate was halved over three years in programme areas, which is faster than the national trend. We recommend addressing socioeconomic inequalities when tackling stunting and providing targeted interventions to mothers during the early weaning period.

## 1. Introduction

Undernutrition remains a major health problem in low-and-middle-income countries despite considerable progress in reducing stunting rates among under-five children globally. Undernutrition has severe consequences for children as it can affect their physical growth, making them prone to infectious diseases and increasing the risk of morbidity and mortality from infectious diseases [1]. If it persists, it can affect children’s mental development and consequently have a negative impact on learning and limit the socioeconomic potential of the children affected [1].

Overall, undernutrition prevalence rates in Bangladesh have shown a declining trend over the decades [2]. The prevalence of stunting, where a child is short for their age, consistently decreased from 43% in 2007 to 31% in 2017–2018 [3]. Also, underweight, i.e., low weight for age, prevalence declined from 41% to 22% and wasting, i.e., low weight for height, prevalence declined from 17% to 8% from 2007 to 2017–2018 [3]. The prevalence of stunting varies between rural and urban areas (33% vs. 25%) and across the country [3]. According to the 2017–2018 Bangladesh Demographic Health Survey (BDHS), among the eight divisions, the prevalence of stunting was the highest in Sylhet (43%), followed by Mymensingh (36%), Barishal (33%), Chattagram (33%), Rajshahi (31%), and Rangpur (30%). It was the lowest in Khulna and Dhaka (26%) [3].

The global target set by the World Health Organisation (WHO) in 2012 for reducing stunting in children under five was 40% by 2025 [4,5]. To meet the target, countries need to achieve an Annual Average Rate of Reduction (AARR) of 3.9% in stunting [4,5]. However, the AARR of stunting in Bangladesh is only 2.7%, highlighting the need for upscaling interventions to tackle stunting in children [4,5].

As the causes of undernutrition are multi-factorial [6], the WHO has recommended a multi-sectoral approach to tackle the problem. It includes health, education, water and sanitation, socioeconomic conditions, agricultural sectors, food security, and poverty alleviation with the involvement of all levels of stakeholders and ensuring good governance, ensuring accountability, and reducing socioeconomic inequalities [7]. These include nutrition-specific measures such as the promotion of breastfeeding, iron and folic acid supplementation, multi-micronutrient supplementation, Vitamin A supplementation, and nutrition-sensitive measures such as preventing early marriage, promoting the completion of secondary education, improving the socioeconomic status, and increasing access to safe water and sanitation [7].

Bangladesh has implemented a series of programmes to tackle the problem of child undernutrition. Essential nutrition interventions were scaled-up in the Health, Population, and Nutritional Sector Development Programme from 2011 to 2016 [8]. In addition, the National Nutrition Service operational plan was approved in 2011, which ensured the involvement of other key sectors such as the agriculture, food, and industry sectors. In accordance with the National Nutrition Policy 2015, the Second National Plan of Action for Nutrition (2016–2025) is currently being implemented. It focuses on the following six thematic areas: (1) nutrition for all, adopting a lifecycle approach, (2) agriculture, (3) social protection, such as food and cash transfers for vulnerable people, (4) integrated and comprehensive social and behaviour change communication, such as promoting a balanced diet and breastfeeding for infants and young children, (5) monitoring and evaluation to enable evidence-based decision making, and (6) capacity building. Complementary to the public sector, civil society organisations play a large role in tackling child undernutrition in Bangladesh [7]. 

Max Foundation is an international non-governmental organisation that has been working in Bangladesh since 2006 [9]. It operates in 62 union Parishads within five districts in southern coastal Bangladesh (Barguna, Patuakhali, Jessore, Khulna, and Satkhira), in which the intervention reached 1.2 million people, primarily targeting hard-to-reach households [9,10]. It initially implemented a Water, Sanitation, and Hygiene (WASH) programme, but, driven by scientific evidence, it changed its operating model in 2018 and created an integrated programme, Healthy Village, which combines WASH with nutrition and care to holistically address the many drivers of stunting in children under five. In addition, it introduced an innovative business model through the engagement of local entrepreneurs and social enterprises to support the Healthy Village approach and ensure long-term sustainability. The programme’s ultimate goal is to provide a healthy start in life to as many children as possible in the most effective and sustainable way [9]. The programme is set to achieve zero stunting in programme areas by 2030 [9].

Under the WASH component of the programme, trained staff provide sanitation education, particularly handwashing practices, in both the community and schools. The programme also advocates for investing in sanitation facilities and engages local governments to subsidise latrine purchases for the poorest households [9]. Additionally, the programme promotes demands on WASH facilities by enhancing sanitation product marketing and strengthening the business model for local entrepreneurs selling sanitation products [10]. Through an innovative approach, it supports SaniMarts (sanitary latrine producers, water and sanitation product sellers, and sweepers) in understanding customer needs and increasing the demand for products and customer outreach via health-promoting agents (HPAs) [10]. The programme trains selected women in program areas to become HPAs and micro-entrepreneurs, thereby increasing community access to quality sanitation products and providing income-generating opportunities to the local community [10]. In the programme areas, 186 local women have been trained to become HPAs and another 186 have become sanitation entrepreneurs [10].

Under the nutrition component of the programme, trained community health promoters provide health education on exclusive breastfeeding and child nutrition and demonstrate healthy meal preparation [9]. Moreover, the programme provides growth monitoring charts to communities and trained community health promoters conduct quarterly child growth monitoring sessions. These sessions also include group discussions among mothers to share and learn about best practices for child growth [9].

This research aimed to study how the stunting prevalence changed in children in households targeted by the programme. An independent team analysed the data to determine the trend of stunting prevalence in children under five from 2018 to 2021 in the programme areas. Additionally, factors contributing to stunting in the study population were explored.

## 2. Materials and Methods

### 2.1. Study Design and Setting

This is a cross-sectional, secondary data analysis study using data in communities in which Max Foundation has supported interventions. A cohort analysis was also conducted to analyse the change in stunting status over time and to validate the findings from the cross-sectional study. The study setting comprised the programme implementation areas (62 union Parishads within five districts in southern coastal Bangladesh (Barguna, Patuakhali, Jessore, Khulna, and Satkhira)).

### 2.2. Data Sources 

Three different data sets were used in the study. 

A household census: The 2018 household census collected information on household characteristics, the availability and accessibility of safe WASH facilities, household investment in WASH, and the household practice of monitoring under-five child growth. The census covered the households in programme areas and the response rate was reported to be 99%.

The questionnaire for the household survey was developed internally by Max Foundation. Max Foundation’s partner organisations administered the survey. Data collection training was provided to the core team, which was cascaded down to the field staff. Data were collected using tablets during face-to-face interviews. After submitting the data to the server, a 2–5% random sample of the data was selected and key questions were verified through a short phone-based survey by call centre staff. 

Anthropometric measurements: Anthropometric measurements (height and weight) of children were taken quarterly from October 2018 to December 2021 in the programme areas and were used to calculate height-for-age z-scores using the WHO’s anthro package in R software accessed via https://www.who.int/tools/child-growth-standards/software (accessed on 1 December 2022) [11].

Child measurements were taken by partner organisations using measurement equipment that adhered to WHO guidelines. In 2018, they received a 1-day training course on how to take child measurements, depending on the child’s age. A refresher training course was given in 2020. All child measurements were taken during courtyard sessions where multiple children were measured, followed by the staff discussing key messages around child health with the children’s caregivers. 

The data set contains measurements and z-scores of children from October 2018 to December 2021. For this study, data were cleaned by removing any biologically implausible z-scores according to WHO standards and extreme values (*n* = 57,346) and measurements of children above five years of age (*n* = 8248). This accounted for 11.6% of the total number of measurements. The cleaned data set contained 497,729 measurements of 74,535 under-five children.

Nutrition and child feeding practices: The 2018 nutrition survey covered 24 unions within five districts. The data set contained information on child feeding practices, exclusive breastfeeding, and acute malnutrition status for children under five. The survey areas were selected using random sampling and target respondents were randomly selected mothers of under-five children (focusing on children aged between 24 and 59 months).

A summary of the data sets is presented in Appendix A. The Max Foundation survey data for each survey (the data used for this study) can be found at https://www.kaggle.com/datasets?search=Max+Foundation+Bangladesh+, accessed on 1 October 2022.

### 2.3. Data Selection 

A detailed data selection flowchart is provided in Figure 1.

For risk factor analysis, the household census data set was paired with the anthropometric data set by the household identification number. However, the nutrition data set could not be paired with the anthropometric data set due to the absence of a common identification variable in the two data sets. Thus, the risk factor analysis was limited to the WASH variables included in the household census. 

### 2.4. Study Sample Size 

For point-prevalence estimates of childhood stunting, anthropometric measurements from Quarter 4 (October–December) from all four years were chosen (*n* = 132,038). Quarter 4 was selected as the growth measurements for 2018 were mainly available from Quarter 4. From the total of 132,038 measurements, under-five children with trackable continuous serial growth measurements for four years starting from Quarter 4, 2018 were selected (*n* = 3448) for a more in-depth cohort study.

Out of the 283,844 households included in the household census, we selected households with at least one under-five child (*n* = 82,864) for pairing with children who had anthropometric data from Quarter 4, 2018 (*n* = 26,585). Pairing households and anthropometric data sets resulted in a total of 20,174 under-five children, all of which were included in the risk factor analysis.

### 2.5. Study Variables

The outcome variable was stunting (low height for age) as a binary variable. Stunting is defined by the WHO classification of children as a height-for-age z-score of more than two standard deviations below the WHO reference population median value [2].

The covariates included in the analyses were as follows: (1) child age group, (2) child gender, (3) residency areas (new areas where the programme’s implementation started in 2018 or follow-up areas where the implementation started early in 2016), (4) mother’s age group, (5) mother’s education status, (6) household size, (7) number of under-five children in a household, (8) self-reported household socioeconomic status as either hardcore-poor, poor, middle-income, or rich, (9) household accessibility of safe water for drinking, (10) household accessibility of safe water for cooking, (11) household accessibility of safe water for washing utensils, (12) household accessibility of safe water for washing children’s clothes, (13) water source, (14) household water availability all year round, (15) self-investment in water sources, (16) household ownership of water sources, (17) distance from the household to water sources, (18) household accessibility of latrines, (19) latrine type, (20) self-investment in latrines, (21) ownership of latrines, (22) distance from the household to a latrine, and (23) monitoring of child growth by the household. Covariates were coded as either binary or categorical variables. The descriptive statistics of the covariates are presented as frequencies and percentages.

### 2.6. Statistical Analyses

Descriptive statistics are provided for the point-prevalence of stunting in children under five in programme areas from 2018 to 2021, Quarter 4. 

For a cohort of under-five children who received continuous growth measurements from 2018 to 2021, the McNemar test was used to explore the change in stunting prevalence (new cases and cases recovered from stunting) in the same group of children over the time period. This was done to validate the findings from the cross-sectional study. A *p* value < 0.05 was considered statistically significant.

Univariate analyses were applied to explore the relationship between stunting and each risk factor, one at a time. A forward selection method was used to select variables for the multivariate model using a *p*-value cut-off of 0.05 [12]. Covariates with a *p* value of <0.05 in the final model were considered statistically significant predictors of childhood stunting.

Cluster analyses by village, union, and district were performed using multilevel generalised linear regression analysis with the binomial family and logit link [13]. The hierarchical variables were the village, union, and district that the children resided in. All covariates tested in the univariate analysis were included for the cluster analysis. The statistical test results are presented as the odds ratio (OR) with the confidence interval (CI) and the *p* value. A *p* value < 0.05 was considered statistically significant. All analyses were conducted in Stata BE (V.17).

### 2.7. Ethical Statement

No ethical approval was needed for this study as the study used anonymised secondary data.

## 3. Results

### 3.1. Point-Prevalence Estimates of Stunting for Children under Five (2018 to 2021, Q 4)

The stunting prevalence for children under five in the programme areas (*n* = 132,038) was 51% in 2018, 53% in 2019, 35% in 2020, and 25% in 2021 (Figure 2). There was a 50% reduction in stunting prevalence from 2018 to 2021. 

The stunting prevalence rates by district from 2018 to 2021 are presented in Appendix A. Among the five districts, Patuakhali had the highest stunting prevalence rate (55% in 2018 and 30% in 2021). 

### 3.2. Change in Stunting Prevalence over Time in a Cohort of Children

In a sub-group of under-five children who each were followed-up continuously for four years (*n* = 3448), 25% of the normal (non-stunted) children in 2018 developed stunting in 2019 (Table 1). The percentage of normal children who developed stunting was reduced to 9% in 2020 and remained constant at 9% in 2021 (Table 1). All of the changes in stunting prevalence between different years were statistically significant, having a *p* value < 0.001. Eighty-nine percent of the under-five stunting cases in 2018 (the baseline) attained normal growth by the end of 2021.

### 3.3. Characteristics of the Study Population in the Risk Factor Analysis

A total of 20,174 children were included in the risk factor analysis. A total of 38% of the children included in the study were less than two years (24 months) of age, and 51% were male.

Information on the children’s household characteristics is presented in Table 2. Most of the mothers of under-five children (62%) in this study were 21–30 years of age, and 42% had not completed secondary education. A total of 62% of the households self-declared as poor or hardcore poor. Although 92% of the households could access a water source all year round, only 18% had their own water sources. A total of 83% had their own latrine in their household, and 54% of the households used improved pit latrines. It was found that 35% to 46% of hardcore-poor or poor households had access to improved latrines compared with 70% of middle-income and 87% of rich households. A total of 92% of the households reported that they did not monitor their child’s growth before the programme offered the service.

Although the nutrition-related factors could not be included in the risk factor analysis, results from the nutrition survey are presented in Appendix A, to provide background information on child feeding practices in some programme areas in 2018. The results show that 77% of the children surveyed were breastfed in the first three days of life. Approximately half of the children (52%) were exclusively breastfed during the first 6 months of life. After six months of life, 38% of the children continued breastfeeding and 40% were given complementary foods. From 12 months of age, approximately 70% said that the child’s diet contained animal-based foods, vegetables, and fruits and that they gave their children food three to four times or more a day.

### 3.4. Factors Associated with Stunting in Children under Five 

The results from the univariate analysis are shown in Appendix A. The multivariate model included the following ten covariates: child age, child gender, residency areas, mother’s education, number of under-five children in a household, household socioeconomic status, household ownership of water sources, latrine type, household access to safe water for cooking, and monitoring of child growth by the household. The multivariate analysis results are shown in Table 3.

Each variable’s OR is the adjusted OR, controlling for all other variables included in the multivariate analysis.

In the final model, the risk of stunting was found to have increased in children aged 1–2 years compared with children under one year of age (aOR: 1.32, 95% CI: 1.20, 1.45), in hardcore-poor (aOR: 1.46, 95% CI: 1.27, 1.68), poor (aOR: 1.50, 95% CI: 1.33, 1.70), and middle-income (aOR: 1.32, 95% CI: 1.16, 1.49) households compared with rich households, and in children with mothers who were illiterate (aOR: 1.25, 95% CI: 1.09, 1.44), could just read and write (aOR: 1.35, 95% CI: 1.16, 1.56), had completed primary education (aOR: 1.19, 95% CI: 1.07, 1.33), or had completed secondary education (aOR: 1.20, 95% CI: 1.08, 1.34) compared with children with mothers who had completed higher education (*p* value < 0.05).

In addition, the odds of stunting significantly increased in children whose growth was not monitored or measured by households compared with those who had (aOR: 1.23, 95% CI: 1.11, 1.37). Being a male child (aOR: 1.10, 95% CI: 1.10, 1.26), residing in follow-up areas where the programme’s implementation started in early 2016 (aOR: 1.18, 95% CI: 1.10, 1.26), having more than one under-five child in the household (aOR: 1.14, 95% CI: 1.05, 1.23), lacking access to safe water for cooking compared with those who had access (aOR: 1.14, 95% CI: 1.07, 1.21), using an unimproved latrine (aOR: 1.09, 95% CI: 1.03, 1.16), and using a shared latrine (aOR: 1.07, 95% CI: 0.99, 1.16) were found to significantly increase the odds of childhood stunting in the programme areas.

There was no evidence of cluster effects by village, union, and district on stunting in children under five (Appendix A).

## 4. Discussion

The findings in this study suggest that the prevalence of childhood stunting in the programme areas decreased by 50% from 2018 to 2021. A significant decline in stunting prevalence occurred between 2019 and 2020. The risk factors strongly associated with stunting in this study were lower household socioeconomic status, a mother with a lower education level, child age (one to two years old), households who did not monitor their children’s growth before the programme’s implementation, residing in an area where the programme’s implementation started early, and male gender. Having more than one under-five child in the household, household access to safe water for cooking, using unimproved latrines, and sharing latrines with other households slightly increased the risk of childhood stunting in the programme areas.

Previous studies reported various age groups as predictors of stunting in Bangladesh. Similarly to our finding, two studies reported that children aged one to two years are more likely to be stunted compared with children aged less than one year [6,14]. In contrast, the analysis using data from the BDHS 2014 showed that the probability of stunting increased with age and was the highest in children aged 36–47 months [5]. In another analysis of data from the Multiple Indicator Cluster Survey 2019, the risk of stunting was the highest among children in the 24–36-month age group [14].

The use of different age ranges as the reference group in different studies can explain the conflicting results between those studies. In studies that reported a higher risk of stunting in children more than 24 months of age, the six-month-old age group was used as the reference group in a comparison with the other age groups [5,14]. For studies that reported a higher risk of stunting in children aged one to two years, the less than one year age range was used as a reference group [6,14].

The increased risk of stunting in the one-to-three-year age group can be explained by the risk of inappropriate complementary feeding of children after six months of age or the delayed introduction of complementary foods. This can lead to the faltering of child growth and stunting in later years [6,15].

Of the other key predictors from our study, poor household socioeconomic conditions and a mother with a low education level are commonly reported predictors of childhood stunting in Bangladesh [5,6,14,16,17,18,19].

Poorer households are more likely to have poorer water, sanitation, and hygiene environments than well-off households [20]. This was also true in our study, where hardcore-poor or poor households had less access to improved latrines and used shared water sources more compared with middle-income and rich households. Thus, children from households with low socioeconomic conditions are more prone to the development of common childhood diseases, such as diarrhoea, that can contribute to stunting [20,21,22]. Furthermore, poor households cannot afford to buy nutritious food and have lower household food security, contributing to childhood malnutrition [6,23].

Regarding mothers’ education level, mothers with lower education levels may have less knowledge of complementary feeding, child nutritional requirements [18], and the importance of monitoring child growth, thereby exposing their children to the risk of stunting. In addition, a low education level is correlated with a lower income and lower spending power in the household [14,18]. Spending on child nutrition and health will therefore be lowered, which increases the child’s risk of stunting.

The findings on child gender and number of children in the household are consistent with results from previous studies in Bangladesh, which demonstrated that male children [5] and children in households with many under-five children [17] have a higher risk of stunting.

The higher risk of stunting in boys compared with girls can be explained by differences in biological factors between the two genders. Boys have riskier growth strategies that predispose them to higher preterm birth rates, weaker immune responses, and higher energy needs than girls [24]. Additionally, mothers of male children were found to perceive their sons to be hungrier than their daughters and be more likely to introduce complementary foods earlier than recommended [24].

The increased risk of stunting in children living in households with multiple children under five can be attributed to inappropriate complementary feeding practices, which are prevalent in families with short birth spacing practices [25]. Moreover, the presence of many young children in one household can strain resources, such as time and money, resulting in less individualised care for each child.

Another finding from this study is that children who did not receive growth monitoring by households before the programme’s implementation had a higher risk of stunting than those who received growth measurements. Previous studies examining the link between growth monitoring and childhood stunting showed conflicting results [26,27]. However, when coupled with maternal nutrition counseling, child growth monitoring practices have been shown to reduce malnutrition [26]. 

Our study findings on the use of unimproved latrines as a risk factor for stunting are supported by a similar study using the BDHS data, where the odds of stunting were found to be lower among households who had improved toilets compared with those who did not have them (OR: 0.92; CI: 0.84, 1.06) [18]. Also, a risk factor analysis of childhood stunting in 137 developing countries showed that environmental risk factors (water and sanitation) play a large role in childhood malnutrition, followed by maternal and child nutrition and infection in South Asia [28]. This could explain the slight increase in the risk of stunting in children living in households with poor access to safe water for cooking.

The initial stunting prevalence rates in the programme areas were above the national prevalence rate of 31% in 2018 [3] and 26% in 2019 [5,6,14,19]. The higher rate of stunting among under-five children in the program areas compared with the national rate at the 2018 baseline can be attributed to differences in the sampling design and target population between the BDHS and Healthy Village programmes. The BDHS utilises population-representative data obtained through a two-stage probability sampling process drawn from the census frame, and anthropometric measurements were taken from the children in selected households [3]. In contrast, the Healthy Village programme collected program-specific data through convenient sampling, where trained staff took serial anthropometric measurements during courtyard sessions. Since the programme was implemented in rural areas and aims to reach vulnerable households, the measurements are likely to have been taken from vulnerable households where most children with stunting live. Additionally, staff may have been more likely to under-report children with normal growth as they focused on detecting children with stunting.

In line with the national trend, the prevalence of stunting in the programme areas declined from 51% in 2018 to 25% in 2021. The dramatic decline in the stunting rate over time in the programme areas could be due to a combination of numerous factors. 

In the programme areas, there are 248 community health promoters trained by Max Foundation to raise community awareness about child growth monitoring, exclusive breastfeeding, and child feeding practices and measure child growth quarterly, starting in 2018. Thus, the community living in the programme areas might have become more aware of the importance of child growth monitoring and appropriate child feeding practices and more likely to monitor their child’s growth, which may have contributed to the reduction in the stunting rate in programme areas. By the end of 2021, local governments had declared 409 out of the 1674 villages in the programme areas to be Healthy Villages since they met the targets set on the 18 WASH and child nutrition indicators. Thus, the stunting rate reduction found in this study might be attributable to the improved WASH and child nutrition conditions in the programme areas. 

Our study has shown that the risk of stunting had increased in children residing in areas where the programme was initiated in early 2016 compared with areas where the programme was implemented later. This implies that the programme reached the more vulnerable areas as intended since its early implementation. This might have contributed to the rapid reduction in stunting in the programme areas.

Our study has shown that household socioeconomic conditions, the mother’s education level, and child age (one to two years) are the strongest determinants of the nutritional status of children in households, including the risk of stunting. Given the short observation period, it is less likely that changes in socioeconomic and education factors were the main contributors to the reduction in stunting. That change can possibly be explained by the fact that the intervention implemented had adopted an integrated approach, simultaneously targeting other modifiable factors such as community knowledge and awareness, growth monitoring, and access to and the use of improved sanitation measures. The programme also had a strong component of health education on the proper feeding of children, a fact that may well weigh heavily as an explanation for the positive results documented. 

Our study results can inform future programme planning. Childhood stunting is a complex issue with multiple risk factors, and addressing childhood stunting necessitates addressing social and economic inequalities. Moreover, expanding the local entrepreneurship model will allow the communities in the programme areas to generate extra income and improve the household socioeconomic conditions in the long term, which may contribute to a further reduction in stunting. Lastly, it is important for community health promotors in the programme areas to provide targeted health education sessions and conduct frequent follow-ups with weaning mothers to ensure that children receive the appropriate complementary foods in a timely manner. 

This study has some strengths. Firstly, the analysis was based on a large number of serial growth measurements of children, allowing us to detect meaningful differences in childhood stunting prevalence across the years. Secondly, the risk factor analysis was based on a household census, which had a large sample size and a 99% response rate. The large sample sizes increase the generalisability of the study results to children residing in Barguna, Patuakhali, Jessore, Khulna, and Satkhira in Bangladesh. 

However, this study has limitations. Due to the unavailability of data, other possible important predictors of stunting, such as the child’s size at birth (fetal growth restriction) [28], non-exclusive breastfeeding practices [28,29], the nutritional parenting of families [30], infection [28,31], birth order, place of birth (health facility or home delivery), household religion, healthcare service uptake, such as BCG vaccine uptake, and frequency of antenatal care visits by mothers during pregnancy [16], could not be explored in this study. Moreover, implausible z-scores and height measurements were excluded from the analysis. When analysing the excluded data, the implausible z-scores, which were mostly from measurement errors, were higher in children aged one to two years old compared with other age groups. This warrants careful interpretation of the high risk of stunting in the one-to-two-year age group found in this study.

The risk factor analysis was only performed using data from a baseline survey in 2018. Including nutrition-related factors and the longitudinal changes in baseline variables could deepen the analysis and enhance the comprehensiveness of the results. Lastly, due to the cross-sectional study design, we could only determine the association and not a causal link between the risk factors and stunting.

## 5. Conclusions

The stunting rate was found to have halved over 3 years in programme areas, which is higher than the national trend. The factors with the highest risk for stunting were found to be poor household socioeconomic conditions, a low level of maternal education, and child age (one to two years). It is crucial to tackle socioeconomic inequalities when addressing stunting and provide targeted interventions to mothers weaning their young children.

## Figures and Tables

**Figure 1 children-11-00650-f001:**
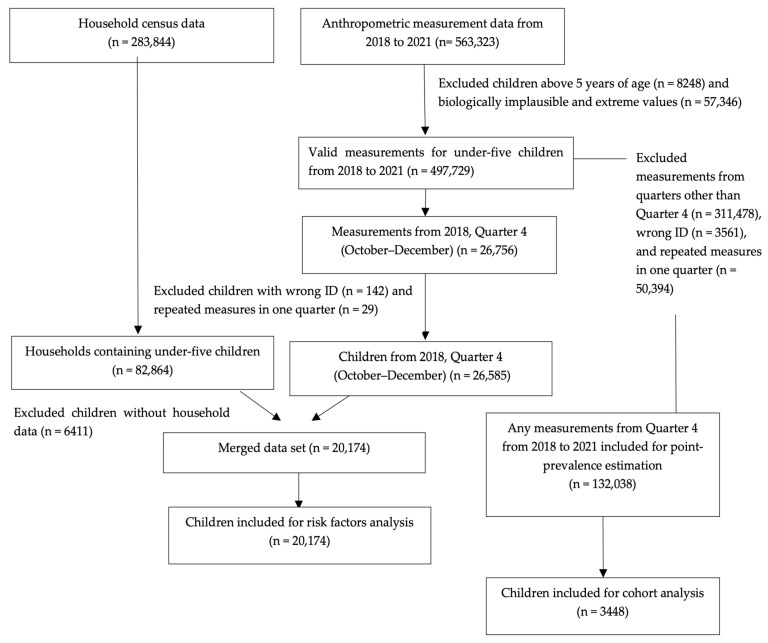
Flowchart showing the cleaning and selection of data from the household census and anthropometric measurement data sets for childhood stunting prevalence and risk factor analysis.

**Figure 2 children-11-00650-f002:**
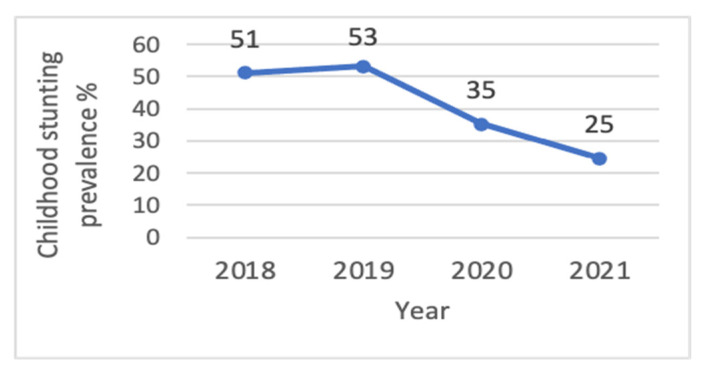
Point-prevalence of childhood stunting in programme areas in five districts of southern coastal Bangladesh from 2018 to 2021, Quarter 4 period (October–December) (*n* = 132,038).

**Table 1 children-11-00650-t001:** Change in childhood stunting prevalence over time from 2018 to 2021 in programme areas in five districts of southern coastal Bangladesh.

	Percent (%) Change in Childhood Stunting Prevalence between Different Years					
	2018 and 2019	2018 and 2020	2018 and 2021	2019 and 2020	2019 and 2021	2020 and 2021
Description	(*n* = 3448)	(*n* = 3448)	(*n* = 3448)	(*n* = 3448)	(*n* = 3448)	(*n* = 3448)
Normal to stunting	25%	15%	11%	9%	7%	9%
Stunting to normal	11%	31%	38%	39%	49%	19%
*p* value for McNemar †	<0.001	<0.001	<0.001	<0.001	<0.001	<0.001

† McNemar test to assess the changes in stunting prevalence between different years.

**Table 2 children-11-00650-t002:** Characteristics of the study population and household information (*n* = 20,174).

Variables	Frequency	Percent (%)
District (*n* = 20,174)		
Barguna	421	2
Jessore	1961	10
Khulna	4567	23
Patuakhali	7979	39
Satkhira	5246	26
Programme areas (*n* = 20,174)		
Follow up (implementation started early in 2016)	5626	28
New area (implementation started in 2018)	14,548	72
Child age group (*n* = 20,174)		
0 to 1 y	2843	14
1 to 2 y	4967	24
2 to 3 y	5022	25
3 to 4 y	4393	22
4 to 5 y	2949	15
Child gender (*n* = 20,174)		
Male	10,216	51
Female	9958	49
Yes	10,341	51
Mother’s age group (*n* = 20,174)		
Age ≤ 20	2337	12
Age 21–30	12,527	62
Age 31–40	3899	19
Age > 40	1410	7
Mother’s education status (*n* = 20,174)		
Illiterate	1735	8
Read and Write	1437	7
Primary	7389	37
Secondary	7789	39
Higher	1823	9
Household size (*n* = 20,174)		
Size ≤ 4	8126	40
Size 5 to 7	9987	50
Size > 7	2061	10
Number of under-five children in a household (*n* = 20,174)		
One under-five child per household	17,356	86
More than one under-five child per household	2818	14
Household socioeconomic status (*n* = 20,174)		
Hardcore poor	2949	15
Poor	9600	48
Middle-income	6341	31
Rich	1284	6
Household access to safe water for drinking (*n* = 20,174)		
No	37	0.2
Yes	20,137	99.8
Household access to safe water for cooking (*n* = 20,174)		
No	12,290	61
Yes	7884	39
Household access to safe water for washing utensils (*n* = 20,174)		
No	14,668	73
Yes	5506	27
Household acces to safe water for washing children’s clothes (*n* = 20,174)		
No	16,639	83
Yes	3535	18
Water source (*n* = 20,174)		
Tubewell (Deep)	12,142	60
Others (shallow tubewell, treated pond water, etc.)	8032	40
Household water availability all year round (*n* = 20,174)		
No	1534	8
Yes	18,640	92
Self-investment in water sources (*n* = 20,174)		
No	13,997	69
Yes	6177	31
Water source ownership (*n* = 20,174)		
Shared or owned by others	16,498	82
Self-owned	3676	18
Distance from household to water source (*n* = 20,174)		
>30 min or no access at all	4420	22
<30 min	15,754	78
Household access to improved sanitation (*n* = 20,174)		
No	200	1
Yes	19,974	99
Type of latrine (*n* = 20,174)		
Unimproved pit latrine/hanging/open defecation	9188	46
Improved pit latrine	10,986	54
Self-investment in latrines (*n* = 20,174)		
No	2534	13
Yes	17,640	87
Latrine ownership (*n* = 20,174)		
Shared	3447	17
Self-owned	16,727	83
Distance from household to latrine (*n* = 20,174)		
More than 12 steps or no access at all	12,664	63
Attached or within 12 steps	7510	37
Monitoring child growth by household (*n* = 20,174)		
No	18,602	92
Yes	1572	8

**Table 3 children-11-00650-t003:** Multivariate logistic regression analysis of risk factors for childhood stunting in programme areas in five districts of southern coastal Bangladesh (*n* = 20,174).

Covariates	aOR	95% CI (aOR)		Standard Error	z	*p* > |z|
		Lower Limit	Upper Limit			
Child age group						
0–1 year	1.00	Reference group				
1 to 2 years	1.32	1.20	1.45	0.06	5.82	<0.001 *
2 to 3 years	1.00	0.91	1.10	0.05	0.03	0.98
3 to 4 years	1.00	0.91	1.11	0.05	0.10	0.92
4 to 5 years	0.94	0.85	1.04	0.05	−1.16	0.25
Child gender						
Female	1.00	Reference group				
Male	1.10	1.04	1.16	0.03	3.36	0.001 *
Programme areas						
New area	1.00	Reference group				
Follow up	1.18	1.10	1.26	0.04	4.77	<0.001 *
Mother’s education status						
Higher	1.00	Reference group				
Illiterate	1.25	1.09	1.44	0.09	3.17	0.002 *
Read and Write	1.35	1.16	1.56	0.10	4.02	<0.001 *
Primary	1.19	1.07	1.33	0.07	3.19	0.001 *
Secondary	1.20	1.08	1.34	0.06	3.46	0.001 *
Number of under-five children in a household						
One under-five child	1.00	Reference group				
More than one under-five child	1.14	1.05	1.23	0.05	3.07	0.002 *
Household socioeconomic status						
Rich	1.00	Reference group				
Hardcore poor	1.46	1.27	1.68	0.10	5.31	<0.001 *
Poor	1.50	1.33	1.70	0.10	6.45	<0.001 *
Middle-income	1.32	1.16	1.49	0.08	4.36	<0.001 *
Household access to safe water for cooking						
Yes	1.00	Reference group				
No	1.14	1.07	1.21	0.04	4.10	<0.001 *
Water source ownership						
Self-owned	1.00	Reference group				
Shared or owned by others	1.07	0.99	1.16	0.04	1.79	0.07
Type of latrine						
Improved pit latrine	1.00	Reference group				
Unimproved pit latrine/hanging/open defecation	1.09	1.03	1.16	0.03	2.87	0.004 *
Monitoring of child growth by the household						
Yes	1.00	Reference group				
No	1.23	1.11	1.37	0.07	3.87	<0.001 *

* *p* < 0.05. aOR, adjusted odds ratio; 95% CI, 95% confidence interval. Each variable’s OR is the adjusted OR, controlling for all other variables included in the multivariate analysis.

## Data Availability

The Max Foundation survey data for each survey (the data used for this study) can be found at https://www.kaggle.com/datasets?search=Max+Foundation+Bangladesh+, accessed on 1 October 2022.

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
