# Peer review of "Assessment of Childhood Stunting Prevalence over Time and Risk Factors of Stunting in the Healthy Village Programme Areas in Bangladesh"

_children, 2024, doi:10.3390/children11060650_

Round 1

Reviewer 1 Report

Comments and Suggestions for Authors

Q1:NIPORT: The prevalence of stunting, where a child is short for age, consistently decreased from 43% in 2007 to 31% in 2017-18

But in your Fig2, your data are more than 50%. Why are the differences?

Q2: The increased risk of stunting in the one to three years age group can be explained by the risk of inappropriate complementary feeding to children after six months of age or delayed introduction of complementary foods. That can lead to child growth faltering and stunting in later years

What are the underlying reasons for the guardians to feed children with inappropriate complementary feeding?

Q3: The poorer households are more likely to have poor water, sanitation and hygiene 404 environments than the well-off households [20]. This is also true in our study, where hard- 405 core poor or poor households had less access to improved latrines and used shared water 406 sources more compared to middle-income and rich households. Thus, children from 407 households with low socioeconomic conditions are more prone to develop common child- 408 hood diseases, such as diarrhoea, that can contribute to stunting [20–22]. Furthermore, 409 poor households have lower affordability to buy nutritious food and have lower house- 410 hold food security, contributing to childhood malnutrition [6,23].

Do you think that religion or other factors which may lead to the childhood stunting

Q4: Findings on child gender and the number of children in the household are consistent with results from previous studies in Bangladesh, which demonstrated that male children [5] and children in households with many under-five children [17] have a higher risk of stunting.

Why?

Q5: The dramatic decline in stunting rate over time in the programme areas can be due to a combination of numerous factors.

What are those factors? mothers' education, household socioeconomic conditions and child age?

Q6:Did you consider the genetic factors, constitutional growth may partly contribute to the childhood stunting? How did you rule out those factors in your subjects in your study?

Reviewer 2 Report

Comments and Suggestions for Authors

I think this paper needs much greater clarity.

What is the main aim – the title implies the change in prevalence of stunting – but most of the analyses are devoted to risk factor analysis.

1.      Just focussing on the change in undernutrition between 2018 and 2021why just focus on stunting – what about underweight and wasting – why not look at the changes in composite index of anthropometric failure i.e.. Combine all three -so did wasting and underweight also fall concomitantly?

2.      You have information on the same children so why not treat HAZ, WAZ and WHZ as continuous characters and use a repeated measures ANOVA to look at the changes over the 4 years? This could replace the risk factor analysis which only looks at 2018 and you could test which of the variables is the best predictor.

3.      The number of implausible height-for age z-scores is very high at over 10% which is of concern – the implication is that either the age is incorrect or the measurement was erroneous.  Were the implausible scores equally distributed across all ages?

4.      How were within household effects controlled for as more than 1 child per households was analysed?

5.      Using 0.25 and 0.1 as cut-offs for inclusion/exclusion seems unjustified and a p value of 0.05 should be used. Why not replace all three models with a forward stepwise regression which will show which is the most significant predictor of stunting  etc – with 0.05 required for inclusion.  This analysis should be repeated for years 2019, 2020 and 2021 so one can see if the same variables predict stunting in all 4 years rather than just in 2018. Alternatively use all 4 years together and add in interaction effects of year X variable.

6.      Further clarity of the numbers would help – for example the cohort sample of 3448  - does that refer to only those children who have all 4 measurements?  It seems a very small number given the total sample size.

7.      The point prevalence results and the cohort data are not in complete agreement.  Between 2018 and 2019 point prevalence worsened by 2% from 51% to 53% while the cohort data show an increase in stunting of 14%.  That is why I suggested using a repeated measures ANOVA on the continuous data.

Round 2

Reviewer 2 Report

Comments and Suggestions for Authors

 I would change the p values in table 3 from 0.00 to more exact values e.g. <0.001,0.003 etc.

Author Response

Dear Editor and Reviewers,

We would like to thank you for the opportunity to resubmit a revised copy of this manuscript and express our thanks to the reviewers for their helpful comments and corrections, which we believe have improved the manuscript.

The manuscript has been revised in the following two areas:

  • the abstract, where the aOR and 95% CI were updated as per the revised analysis and
  • the result section, Table 3, where the p values smaller than 0.001 are now reported as p < 0.001 and p values between 0.01 and 0.001 are reported to three decimal places.

The changes made are yellow highlighted in the revised manuscript.